# Research on Secure State Estimation and Recovery Control for CPS under Stealthy Attacks

**Biao Yang** **, Liang Xin and Zhiqiang Long** *

College of Intelligence Science and Technology, National University of Defense Technology,
Changsha 410073, China; yangbiao16@nudt.edu.cn (B.Y.); xinliang@nudt.edu.cn (L.X.)
* Correspondence: zhqlong@nudt.edu.cn

**Abstract:** As the application of cyber-physical systems (CPSs) becomes more and more widespread, its security is becoming a focus of attention. Currently, there has been much research on the security defense of the physical layer of the CPS. However, most of the research only focuses on one of the aspects, for example, attack detection, security state estimation, or recovery control. Obviously, the effectiveness of security defense targeting only one aspect is limited. Therefore, in this paper, a set of security defense processes is proposed for the case that a CPS containing multiple sensors is subject to three kinds of stealthy attacks (i.e., zero-dynamics attack, covert attack, and replay attack). Firstly, the existing attack detection method based on improved residuals is used to detect stealthy attacks. Secondly, based on the detection results, an optimal state estimation method based on improved Kalman filtering is proposed to estimate the actual state of the system. Then, based on the optimal state, internal model control (IMC) is introduced to complete the recovery control of the system. Finally, the proposed methods are integrated to give a complete security defense process, and the simulation is verified for three kinds of stealthy attacks. The simulation results show that the proposed methods are effective.

**Keywords:** cyber-physical systems (CPSs); secure state estimation; recovery control; stealthy attacks; improved Kalman filter; internal model control (IMC)

## 1. Introduction

Cyber-physical systems (CPSs) refer to the integration of cyber components (computing and communication) and physical components (sensors and actuators) that interact in a feedback loop, allowing for potential human intervention, interaction, and exploitation [1]. The widespread utilization of CPSs in diverse sectors such as power transmission, healthcare, communication, military, transportation, automotive, entertainment, and others has a direct and profound impact on everyday life, underscoring their immense importance [2,3]. Nonetheless, the growing integration of the cyber and physical domains has given rise to novel and perilous security challenges. Notably, previous occurrences have illustrated the susceptibility of CPSs. The Sapphire worm, which emerged in 2003, resulted in substantial disruption to websites and internet services [4]. In the year 2010, the Stuxnet virus was specifically designed to attack Iran's nuclear power plant, causing significant damage to its centrifuges and resulting in the reactor being non-functional for a prolonged duration [5]. Similarly, the WannaCry ransomware attack on the National Health Service in 2017 had severe consequences in terms of both human casualties and financial losses [6]. These instances highlight the potential ramifications of attacks on CPSs, which can significantly impact individuals' lives. Consequently, the study of CPS security holds significant importance.

The security defense challenges in CPSs can be classified into two distinct layers: the cyber layer and the physical layer. The defense mechanisms employed in the cyber layer bear a striking resemblance to those utilized in safeguarding information networks.

For example, safeguarding against network attacks can be accomplished by employing encryption keys [7], detecting watermarks [8], and imposing access rights restrictions [9]. The tight integration of CPSs with physical devices introduces a distinctive aspect wherein attacks on the cyber layer can extend their impact to the physical processes themselves. This phenomenon is particularly conspicuous in the realm of automatic control systems, resulting in compromised control performance. Consequently, the protection of the physical layer assumes paramount importance due to its role as the final barrier of defense and its ability to offer substantial defensive benefits.

The security defense problem for the physical layer of CPS can be divided into several stages: attack detection, secure state estimation, and secure control [10]. Among them, each of them has a different and necessary function. However, most of the studies focus on one of these phases for analysis and discussion only.

To tackle the issue of detecting attacks, a proposed solution in [11] suggests a blended detection approach. This approach combines two established detection methods to effectively identify a broad spectrum of false data injection attacks. In a recent study [12], a novel strategy for moving target defense is introduced. This strategy involves the integration of random and time-varying parameters into the control system, with the aim of obstructing attackers from devising stealthy attack sequences. Moreover, a novel control architecture is presented in reference [13], which employs watermark signals and auxiliary systems to identify spoofing attacks that have an adverse effect on the network control system, while simultaneously maintaining control performance. Another method proposed in reference [14] introduces an improved residual-based detection technique that is specifically tailored for the identification of stealthy attacks on omnidirectional mobile robots (OMRs). Additionally, there has been significant research conducted on observer-based detection methods for identifying integrity attacks, including replay attacks, zero-dynamics attacks, covert attacks, and others [15–17].

Various methods have been proposed to ensure secure state estimation in CPSs. One method that can be employed to address switching signals and spurious measurement attacks, and to ensure resilient state estimation, is the utilization of a Bayesian approach based on random sets [18]. In reference [19], a robust asymptotic fault estimation technique is developed specifically for CPSs that experience sensor faults. This technique enables accurate estimation of the system state. Addressing the problem of secure reconstruction in linear CPSs that encounter sparse attacks on both actuators and sensors, a descriptor-switched sliding mode observer is proposed in [20]. This observer is designed to effectively reconstruct the sparse false data injection (FDI) attacks and the system state. In reference [21], a comprehensive solution is presented which combines detection and fusion techniques. This solution is based on the utilization of Kullback–Leibler divergences (KLD) between local posteriors. By employing this approach, the exchange of raw sensor data is eliminated, while simultaneously ensuring secure state estimation. In the study conducted by [22], the distributed dynamic state estimation algorithm was designed. This algorithm utilizes optimal filters and graph theory to obtain local and neighboring gains, thereby improving the accuracy of state estimation. Additionally, a novel approach is presented in [23] for constructing two detection variables that do not rely on the invertibility of the covariance matrix. Subsequently, a multiple information fusion algorithm is developed based on the obtained detection results.

Several approaches have been proposed to ensure secure control in CPS. One potential strategy involves implementing an architectural framework for resilient CPS through the utilization of stochastic model predictive control (MPC). This framework is designed with the objective of attaining robustness in the presence of stochastic uncertainty and ensuring resilient control against cyber attacks [24]. In reference [25], a comprehensive system model incorporating uncertainty is constructed through the analysis of diverse cyber attacks. Subsequently, robust control theory is employed to ensure secure control of the system following an attack. A model predictive switching control strategy is proposed in [26] to address the issue of untrusted data sequences resulting from false data injection attacks.

This strategy is based on attack perception and aims to compensate for the impact of such attacks. The paper [27] centers its attention on the utilization of network-based modeling and proportional–integral (PI) control techniques for a continuous-time direct-drive-wheel system that operates within a wireless network environment. Additionally, the problem of event-triggered synchronization of master–slave neural networks under deception attacks is investigated in [28]. The study proposes appropriate output feedback controllers using the Lyapunov–Krasovskii functional method. Additionally, a novel approach to the analysis of the H∞ performance in discrete-time networked systems, considering network-induced delays and malicious packet dropouts, is presented in reference [29]. The effectiveness of this approach is demonstrated through the application of an inverted pendulum system.

The growing utilization of CPSs has resulted in a heightened emphasis on the security aspects associated with it. Nevertheless, it is worth noting that current research endeavors in the field of CPS security defense tend to focus on isolated facets, thereby neglecting a holistic and all-encompassing approach. In this paper, we examine a particular scenario in which a CPS with multiple sensors is susceptible to stealthy cyber attacks, including zero-dynamics attacks, covert attacks, and replay attacks [30]. To tackle this issue, we present a comprehensive framework for the process of security defense. This framework incorporates an attack detection method that relies on enhanced residuals [31], an optimal state estimation utilizing improved Kalman filtering, and a recovery control strategy based on the optimal state. By integrating these components, a comprehensive security defense process is established, encompassing attack detection and recovery control. The validity of the proposed framework is established through the utilization of design simulations, which demonstrate the feasibility and effectiveness of our method in addressing the security challenges encountered by CPSs. This comprehensive approach addresses the existing research gap by considering the entire security defense process, thereby making a significant contribution to the enhancement of CPS security.

The main contributions of this paper can be summarized as follows.

(1) For the case of attacks on a CPS containing multiple sensors, an optimal state estimation method based on improved Kalman filtering is proposed, which can achieve the estimation of the actual state of the CPS after the attack.
(2) Based on the estimated optimal state, a recovery control strategy is designed. And, combined with the detection method based on improved residuals, a framework for the security defense process is given.

The rest of this paper is organized as follows. In Section 2, the CPS structure and stealthy attacks are described, and the improved residual-based attack detection method is presented. In Section 3, the optimal state estimation based on the improved Kalman filter and the recovery control strategy based on the optimal state are proposed, and a complete framework of the security defense process is given. Simulation experiments are designed in Section 4, and the results are analyzed and illustrated. Section 5 summarizes the full work and provides an outlook for future work.

## 2. Model Building and Detection Methods

### 2.1. System Modelling

The block diagram of the CPS structure with multiple actuators and sensors considering the case of sensor attacks, actuator attacks, and process attacks is shown in Figure 1. The CPS structure consists of two main components: the plant side and the monitoring side. The plant side includes physical devices, actuators, and sensors. The monitoring side utilizes data from measurement channels to remotely control physical devices. The monitoring side controls the actuators by sending control commands to enable desired functions or actions. On the plant side, multiple sensors independently measure the state of the system and transmit this information to the monitoring side, which then generates control commands. Each sensor can provide a partial or complete system state independently. A multi-sensor selection and fusion module is incorporated into the monitoring side to acquire the final system state for generating control commands. This module applies

principles or methods to combine the different system states obtained from sensors. In this CPS structure, attackers primarily target the measurement and actuation channels, as well as directly attack the physical devices, which can have detrimental effects on the CPS.

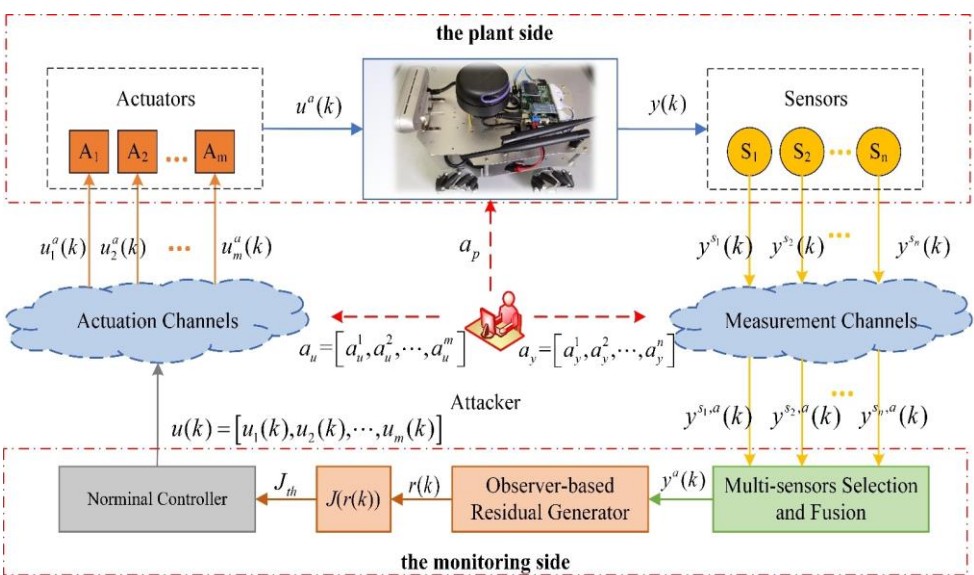

**Figure 1.** Block diagram of CPS structure with multiple actuators and sensors considering sensor attacks, actuator attacks, and process attacks.

The CPS model with multiple sensors, considering the case of containing attacks, is constructed as:

$$
\begin{cases}
x(k+1) = Ax(k) + B(u(k) + a_u(k)) + E_P a_p(k) + w(k), \\
y^{s_i,a}(k) = C^{s_i}x(k) + D(u(k) + a_u(k)) + a_y^i(k) + F_P a_p(k) + v^i(k), \\
y^a(k) = \Phi(y^{s_1,a}|y^{s_2,a}|\cdots|y^{s_n,a}). \, i = 1,2,\ldots,n
\end{cases}
\tag{1}
$$

where $x(k) \in \mathbb{R}^p$ is the state of the system, $u(k) \in \mathbb{R}^m$ is the control command signal output from the monitoring side of the system, $u^a(k) \in \mathbb{R}^m$ ($u^a(k) = u(k) + a_u(k)$) is the control signal input to the plant side after the attack, $y^{s_i,a}(k) \in \mathbb{R}^n$ is the measurement output of the $i$th sensor after the attack, $y^a(k) \in \mathbb{R}^n$ is the system output after multi-sensor selection and fusion, $a_u(k)$ denotes the attack against the actuation channel, $a_p(k)$ denotes the process attack, and $a_y^i(k)$ denotes the attack against the $i$th measurement channel. $\Phi(y^{s_1,a}|y^{s_2,a}|\cdots|y^{s_n,a})$ denotes some multi-sensor fusion method. $E_P$ and $F_P$ are known matrices indicating the locations of components in the system that may be subject to process attacks. $A$, $B$, $C^{s_i}$, and $D$ are the parameter matrices of the system. $C^{s_i}$ denotes the corresponding output matrix when only the $i$th sensor is considered. $w(k)$ denotes the process noise and $v^i(k)$ denotes the measurement noise of the $i$th sensor, both obeying $w(k) \sim N(0, \Sigma_w)$, $v^i(k) \sim N(0, \Sigma_{v^i})$.

**Remark 1.** *Process attacks refer to attacks that are directly applied to physical devices, causing direct damage to the physical devices. On the other hand, actuator attacks and sensor attacks involve actions such as blocking or tampering with signal transmission channels. Given that process attacks directly cause damage to physical devices and may not require further defensive measures, this paper focuses solely on studying the security defense of CPSs when actuator attacks or sensor attacks are present.*

**Remark 2.** *The type of attack mentioned in (1) is of additive attack and does not change the model parameters. Here, a multiplicative attack that causes changes in the model parameters due to functional anomalies of the system, in the process or sensors and actuators due to cyber attacks, is not considered.*

Consider the CPS shown in Figure 1 where the observer-based residual detector is defined at the monitoring side as follows.

$$r_o(k) = y^a(k) - C\hat{x}(k) \tag{2}$$

where $y^a(k) \in \mathbb{R}^m$ denotes the measurement under attack and $\hat{x}(k) \in \mathbb{R}^n$ is the state estimated by the observer. $r_0(k)$ is the residual signal that satisfies $r_0(k) \sim N(0, \Sigma_{r_0})$.

According to the $\chi^2$ test, the residual evaluation function $J(\cdot)$ can be written as

$$J(r_0(k)) = r_0^T(k)\Sigma_{r_0}^{-1}r_0(k) \sim \chi^2(m) \tag{3}$$

So, the logic law of detection to determine the presence of an attack is expressed as

$$\begin{cases} J(r_0(k)) \le J_{th} \Rightarrow attack - free \\ J(r_0(k)) > J_{th} \Rightarrow attacked \end{cases} \tag{4}$$

When the false alarm rate $\alpha$ is given, the threshold $J_{th}$ is set to the upper bound of $\chi_\alpha^2(m)$.

*2.2. Description of Stealthy Attacks*

For integrity stealthy attacks, the following definition is first given.

**Definition 1.** *When there is an attack in the system, given a false alarm rate $\alpha$, and the attack cannot be detected based on the residual signal $r_0(k)$ using the detection logic given in (4), then the attack is said to be stealthy from the detector given in (2).*

According to the definition of stealthy attacks, three types of stealthy attacks are introduced.

(1) Zero-dynamics attack: It requires complete knowledge of the system model to design attack signals against the actuators. It evades the detector of (3) by adding the attack signal $a_u(k)$ to the actuator input without affecting the sensor measurement output, i.e., $a_y(k) = 0$ [32]. Therefore, the attack form can be expressed as $a_u(k) = v^k g$, where the system zero $v$ and the corresponding input zero direction $g$ can be calculated by solving the following equation.

$$\begin{bmatrix} vI - A & -B \\ C & 0 \end{bmatrix} \begin{bmatrix} x_0 \\ g \end{bmatrix} = \begin{bmatrix} 0 \\ 0 \end{bmatrix} \tag{5}$$

where $x_0$ is the initial state of the system for which the input sequence $a_u(k)$ results in an identically zero output.

(2) Covert attack: It also requires complete knowledge of the system model and attacks against both actuation channels and measurement channels. In the actuation channels, the performance of the control system is affected by applying an additive signal $a_u(k)$; however, in the measurement channels, the effect of the input attack on the measurement is eliminated by carefully designing a signal $a_y(k)$ [33]. Given the discrete linear model in (1), $a_y(k)$ can be calculated by the following equation.

$$a_y(k) := -C\sum_{i=0}^{k-1} \left( A^i B a_u(k-1-i) \right) \tag{6}$$

(3) Replay attack: It does not require knowledge of the system model. It only needs to be able to access the signal transmission channels, to attack the control signals, and to record and re-cover the measurement data. The replay attack can be specifically described as [34]: in the measurement channels, the measurement data in the steady-state of the system are recorded in advance, and the actual measure-

ment values are overwritten with the recorded data when the attack is performed (i.e., $y(k) = y(k - \tau), \tau > 0$); while in the actuation channels, $a_u(k)$ is designed to affect the performance of the system. Obviously, the replay attack is stealthy in the steady-state of the system.

**Remark 3.** *Zero-dynamics attack, covert attack, and replay attack are all additive attacks and satisfy the stealthy condition of Definition 1. In addition, zero-dynamics attack and covert attack require complete knowledge of the system model to evade the detection mechanism of (3), while replay attack does not require it and it is stealthy when the system is stable.*

### 2.3. Detection Method

The analysis of stealthy attacks reveals that their core purpose is to attack the actuators to make the control performance by attacking the actuators. On the other hand, attacks against the sensors aim to evade the detection mechanism described in (3). The detection of stealthy attacks can be achieved by exploiting the differences between the states of the system on the monitoring side and the plant side [31]. The detection scheme, as shown in Figure 2, is designed for a system that contains only one sensor capable of independently measuring the full state of the system. The specific working principle is described below.

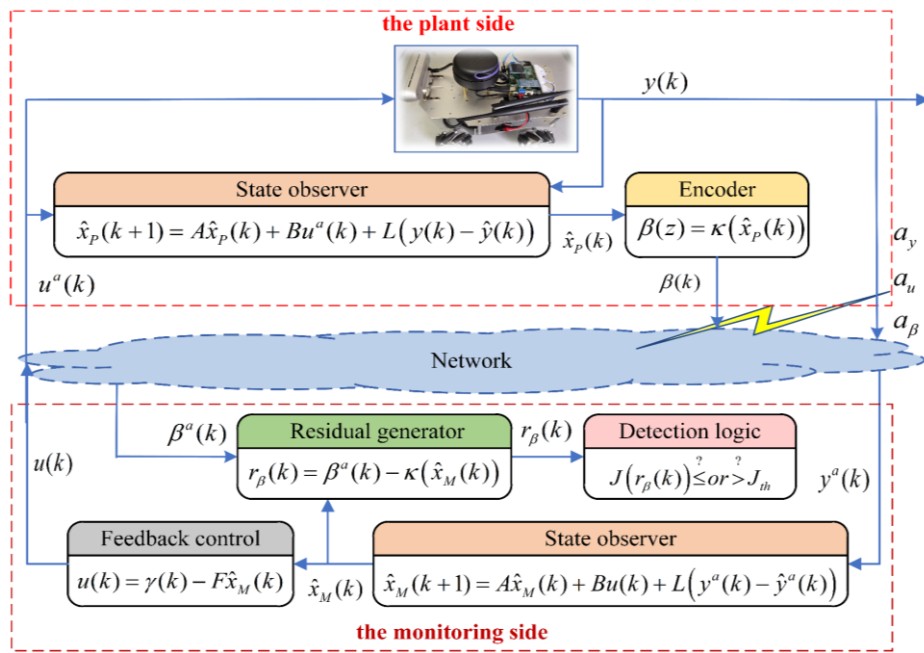

**Figure 2.** The scheme of attack detection based on improved residual.

On the monitoring side, the observer is constructed in the following form:

$$\begin{cases} \hat{x}_M(k+1) = A\hat{x}_M(k) + Bu(k) + L(y^a(k) - \hat{y}^a(k)) \\ \hat{y}^a(k) = C\hat{x}_M(k) + Du(k) \end{cases} \tag{7}$$

On the plant side, the observer is constructed in the following form:

$$\begin{cases} \hat{x}_P(k+1) = A\hat{x}_P(k) + Bu^a(k) + L(y(k) - \hat{y}(k)) \\ \hat{y}(k) = C\hat{x}_P(k) + Du^a(k) \end{cases} \tag{8}$$

From (7) and (8), it can be found that when there is no attack, $\hat{x}_P(k) = \hat{x}_M(k)$; when there is an attack, $\hat{x}_P(k) \neq \hat{x}_M(k)$. This is because the states on both sides are estimated based on different signals. Therefore, this difference can be used for the detection of stealthy

attacks. Also, to avoid the attack when $x_P(k)$ is transmitted to the monitoring side, the following form of the transmission signal is designed:

$$\beta(k) = \kappa(\hat{x}_P(k)) \tag{9}$$

where $\kappa(\cdot)$ indicates a certain encryption policy, which is known only to the system itself.

Therefore, the residual signal used to detect stealthy attacks can be constructed as:

$$r_\beta(k) = \beta(k) - \kappa(\hat{x}_M(k)) \tag{10}$$

## 3. State Estimation and Recovery Control

The CPS is susceptible to a range of stealthy cyber-attacks, which have a substantial impact on its control performance. Undoubtedly, the primary focus for both attackers and defenders lies in determining strategies to enhance the control performance of the control system, thereby achieving their respective attack and defense objectives. Therefore, it is imperative to have a comprehensive controller design solution that is capable of detecting, isolating, and recovering control from stealthy cyber attacks.

Consider a CPS as shown in Figure 1 containing multiple actuators as well as multiple sensors that can be independently measured to obtain the complete system output. The analysis of stealthy attacks in Section 2 shows that the key to their stealthy implementation lies in the tampering with the sensor measurement output, while the attack on the actuators is only to achieve the purpose of the attack and control the specific actions of the system. Therefore, considering $n$ sensors $\Pi_s = \{s_1, s_2, \ldots, s_n\}$, there are unknown sensors suffering from the attack, and the specific attacks are described as follows.

$$y^{s_i,a}(k) = \begin{cases} y^{s_i}(k) \Leftrightarrow attack - free \\ y^{s_i}(k) + a_y^i(k) \Leftrightarrow attacked \end{cases}, i = 1, 2, \cdots, n. \tag{11}$$

Without knowing which sensor is under attack, it is undesirable to directly fuse the sensor output data using the multi-sensor fusion method $\Phi(y^{s_1,a}|y^{s_2,a}|\cdots|y^{s_n,a})$. Therefore, each sensor must be detected in advance and the sensor detected as being under attack must be isolated, and finally, the normal sensors are selected for fusion to obtain the correct system state for recovery control.

### 3.1. Optimal State Estimation Based on Improved Kalman Filtering

To detect whether each sensor is under attack, the detection method described in Section 2 is used to calculate the residual signal for each sensor, allowing for the identification of stealthy attacks, assuming that the residual signals for all sensors are calculated as follows:

$$\Theta = \left\{ r_\beta^{s_1}(k), r_\beta^{s_2}(k), \ldots, r_\beta^{s_n}(k) \right\} \tag{12}$$

According to the $\chi^2$ detection theory [14], drawing on the detection logic of (3) and (4), the detection of stealthy attacks can be achieved for each sensor:

$$\begin{cases} J\left(r_\beta^{s_i}(k)\right) \leq J_{th-\beta} \Rightarrow attack - free \\ J\left(r_\beta^{s_i}(k)\right) > J_{th-\beta} \Rightarrow attacked \end{cases}, i = 1, 2, \cdots, n \tag{13}$$

Assuming that the set of residual signals of the sensors that detected the attack is denoted as

$$\Theta^a = \left\{ r_\beta^{s_i,a}(k) \in \Theta | s_i \in \Pi_s \right\} \subseteq \Theta \tag{14}$$

After detecting the sensors under attack, they need to be isolated and the remaining normal sensors are used to estimate the system state. Therefore, weighting factors are

introduced to achieve isolation and fusion according to (12) and (14). That is, the fusion function $\Phi(y^{s_1,a}|y^{s_2,a}|\cdots|y^{s_n,a})$ in (1) is constructed in the following form:

$$\Phi(y^{s_1,a}|y^{s_2,a}|\cdots|y^{s_n,a}) = \sum_{i=1}^{n} \varepsilon_i y^{s_i,a}(k) \tag{15}$$

where $\varepsilon_i$ is the weight value corresponding to each sensor calculated from the residual signal $r_\beta(k)$, calculated as shown below.

$$\varepsilon_i = \begin{cases} 0 & ,r_\beta^{s_i}(k) \in \Theta^a \\ \left(r_\beta^{s_i}(k)\sum \frac{1}{r_\beta^{s_i}(k)}\right)^{-1} & ,r_\beta^{s_i}(k) \in \Theta - \Theta^a \end{cases} \tag{16}$$

From (16), it can be found that once a sensor detects an attack, its corresponding weighting factor is set to 0, while sensors that are not detecting an attack are weighted and fused according to the corresponding residuals. These residuals provide insights into the extent of state deviation, causing larger residuals to correspond to lower weighting factors during the calculation process.

Then, according to the explanations in Remarks 1 and 2, the model of CPS shown in (1) can be simplified to the following form without considering process attacks:

$$\begin{cases} x(k+1) = Ax(k) + B(u(k) + a_u(k)) + w(k) \\ y^{s_i,a}(k) = C^{s_i}x(k) + D(u(k) + a_u(k)) + a_y^i(k) + v^i(k) \\ y^a(k) = \sum_{i=1}^{n} \varepsilon_i y^{s_i,a}(k) \end{cases} \tag{17}$$

where $y^{s_i,a}(k)$ is the measurement output of the $i$th sensor after attacking and $C^{s_i}$ is the corresponding system output matrix.

Meanwhile, on the monitoring side, the state estimation equation and the observation estimation equation for (17) are:

$$\begin{cases} \hat{x}(k+1|k) = A\hat{x}(k|k) + Bu(k) \\ \hat{y}^{s_i,a}(k+1|k) = C^{s_i}\hat{x}(k+1|k) + Du(k) \\ \hat{y}^a(k+1|k) = \sum_{i=1}^{n} \varepsilon_i \hat{y}^{s_i,a}(k+1|k) \end{cases} \tag{18}$$

where $\hat{y}^{s_i,a}(k+1|k)$ is the value estimated at moment $k$ for the measurement output of the $i$th attacked sensor at moment $k+1$.

Then, the state error covariance matrix $P(k+1|k)$ can be calculated as follows:

$$\begin{aligned} P(k+1|k) &= \text{cov}\{x(k+1) - \hat{x}(k+1|k)\} \\ &= \text{cov}\{Ax(k) + B(u(k) + a_u(k)) + w(k) - A\hat{x}(k|k) - Bu(k)\} \\ &= \text{cov}\{A(x(k) - \hat{x}(k|k)) + Ba_u(k) + w(k)\} \\ &= AP(k|k)A^T + B\Sigma_{a_u}B^T + \Sigma_w \end{aligned} \tag{19}$$

where $\Sigma_{a_u}$ is the covariance matrix of the attack signal $a_u(k)$ and $\Sigma_w$ is the covariance matrix of the process noise $w(k)$.

Similarly, the observation error covariance matrix $S(k+1)$ can be calculated as follows:

$$\begin{aligned} S(k+1) &= \text{cov}\{y^a(k+1) - \hat{y}^a(k+1|k)\} \\ &= \text{cov}\left\{\sum_{i=1}^{n} \varepsilon_i\left(C^{s_i}(x(k+1) - \hat{x}(k+1|k)) + Da_u(k) + a_y^i(k) + v^i(k)\right)\right\} \\ &= \sum_{i=1}^{n} \varepsilon_i^2\left(C^{s_i}P(k+1|k)C^{s_i T} + D\Sigma_{a_u}D^T + \Sigma_{a_y^i} + \Sigma_{v^i}\right) \end{aligned} \tag{20}$$

where $\Sigma_{a_y^i}$ is the covariance matrix of the attack signal $a_y^i(k)$ and $\Sigma_{v^i}$ is the covariance matrix of the measurement noise $v^i(k)$.

The goal of Kalman filtering is for an iterative estimation expression based on errors that can be continuously corrected, in the form shown below:

$$\hat{x}(k+1|k+1) = \hat{x}(k+1|k) + W\widetilde{e}(k+1) \tag{21}$$

where $W$ is the Kalman gain matrix and $\widetilde{e}(k+1)$ is the observation estimation error, defined as

$$\widetilde{e}(k+1) = y^a(k+1) - \hat{y}^a(k+1|k) \tag{22}$$

The solution procedure for $W$ is as follows:

$$
\begin{aligned}
P(k+1|k+1) &= \mathrm{cov}\{x(k+1) - \hat{x}(k+1|k+1)\} \\
&= \mathrm{cov}\{x(k+1) - \hat{x}(k+1|k) - W(y^a(k+1) - \hat{y}^a(k+1|k))\} \\
&= \mathrm{cov}\left\{ \left(I - W\sum_{i=1}^{n}\varepsilon_i C^{s_i}\right)(x(k+1) - \hat{x}(k+1|k)) - W\sum_{i=1}^{n}\varepsilon_i\left(Da_u(k) + a_y^i(k) + v^i(k)\right)\right\} \\
&= \left(I - W\sum_{i=1}^{n}\varepsilon_i C^{s_i}\right)P(k+1|k)\left(I - W\sum_{i=1}^{n}\varepsilon_i C^{s_i}\right)^T \\
&\quad + \sum_{i=1}^{n}\varepsilon_i^2\left(WD\Sigma_{a_u}D^T W^T + W\Sigma_{a_y^i}W^T + W\Sigma_{v^i}W^T\right)
\end{aligned} \tag{23}
$$

Expanding (23) into the form of a trace:

$$
\begin{aligned}
&\mathrm{trace}(P(k+1|k+1)) \\
&= P(k+1|k) - \left(W\sum_{i=1}^{n}\varepsilon_i C^{s_i}\right)P(k+1|k) - P(k+1|k)\left(W\sum_{i=1}^{n}\varepsilon_i C^{s_i}\right)^T \\
&\quad + \left(W\sum_{i=1}^{n}\varepsilon_i C^{s_i}\right)P(k+1|k)\left(W\sum_{i=1}^{n}\varepsilon_i C^{s_i}\right)^T \\
&\quad + \sum_{i=1}^{n}\varepsilon_i^2\left(WD\Sigma_{a_u}D^T W^T + W\Sigma_{a_y^i}W^T + W\Sigma_{v^i}W^T\right)
\end{aligned} \tag{24}
$$

$\mathrm{trace}(P(k+1|k+1))$ to $W$ by taking the partial derivative and making it equal to 0 to compute the optimal $W$:

$$\frac{\partial}{\partial W}\mathrm{trace}(P(k+1|k+1)) = -2P(k+1|k)\left(\sum_{i=1}^{n}\varepsilon_i C^{s_i}\right)^T + 2WS(k+1) = 0 \tag{25}$$

Finally, the optimal $W$ can be obtained:

$$W = \frac{P(k+1|k)\left(\sum_{i=1}^{n}\varepsilon_i C^{s_i}\right)^T}{S(k+1)} \tag{26}$$

Then, substituting (23) into (19) again, the iterative form of the state error covariance matrix can be obtained by simplification:

$$P(k+1|k) = A\left(I - W\sum_{i=1}^{n}\varepsilon_i C^{s_i}\right)P(k|k-1)A^T + B\Sigma_{a_u}B^T + \Sigma_w \tag{27}$$

In conclusion, considering isolation and fusion of the attacked sensors, the optimal state estimation based on the improved Kalman filtering can be summarized as

$$\begin{cases} \hat{x}(k+1|k) = A\hat{x}(k|k) + Bu(k) \\ \hat{x}(k|k) = \left(I - W\sum_{i=1}^{n}\varepsilon_i C^{s_i}\right)\hat{x}(k|k-1) - \sum_{i=1}^{n}\varepsilon_i WDu(k-1) + Wy^a(k) \\ W = P(k|k-1)\left(\sum_{i=1}^{n}\varepsilon_i C^{s_i}\right)^T\left[\sum_{i=1}^{n}\varepsilon_i^2\left(C^{s_i}P(k|k-1)C^{s_i\,T} + D\Sigma_{a_u}D^T + \Sigma_{a_y^i} + \Sigma_{v^i}\right)\right]^{-1} \\ P(k+1|k) = A\left(I - W\sum_{i=1}^{n}\varepsilon_i C^{s_i}\right)P(k|k-1)A^T + B\Sigma_{a_u}B^T + \Sigma_w \end{cases} \tag{28}$$

### 3.2. The Recovery Control Strategy Based on Optimal State

In the context of security defense in CPSs, the goal is to ensure that the system can maintain normal operation even after an attack. Therefore, once an attack is detected using the method described in Section 2 and attack isolation and system state estimation are performed according to the method in Section 3, the next step is to implement recovery control for the system based on the obtained results. This section presents an optimal state-based recovery control strategy and provides a comprehensive overview of attack detection, attack isolation, state estimation, and recovery control. It outlines a complete framework for protecting CPS security, as depicted in Figure 3.

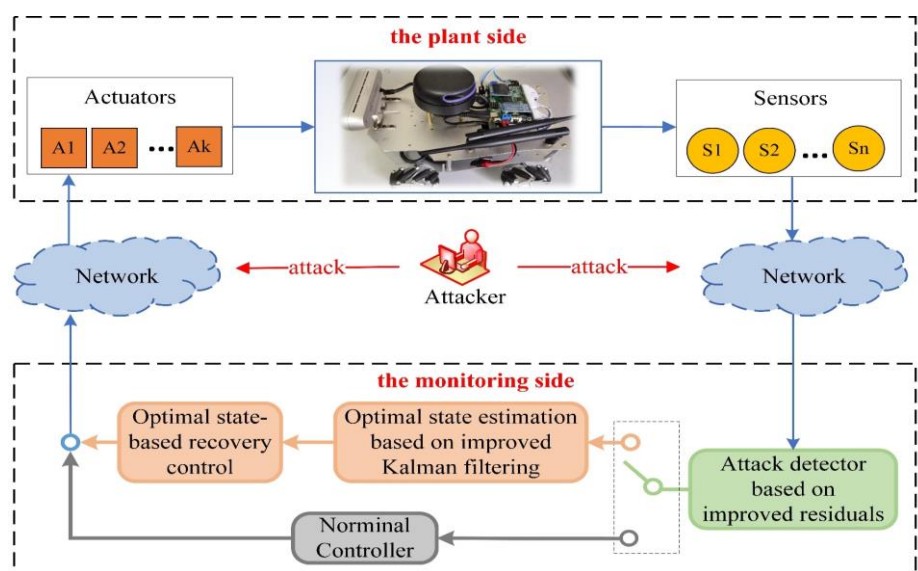

**Figure 3.** The security defense process of CPS includes attack detection, secure state estimation, and recovery control.

In control systems, the primary control objectives are to ensure stable tracking of the reference input and meet the control requirements. The internal model control (IMC) has a wide range of applications in the control field. It enables the system to track the reference input asymptotically with zero steady-state error. This paper proposes a solution to the recovery control problem of CPS under attacks. It introduces an integration link of the reference input signal by combining IMC with the optimal state. This approach aims to achieve stable operation of the attacked system based on the target command.

The reference input signal $\gamma(k)$ is generated by the following model:

$$\begin{cases} \gamma(k) = x_\gamma \\ \dot{\gamma}(k) = 0 \end{cases} \tag{29}$$

Also, the tracking error $e(k)$ is defined:

$$e(k) = y^a(k) - \gamma(k) \tag{30}$$

Then, the state space can be extended as follows:

$$
\begin{pmatrix} e(k+1) \\ z(k+1) \end{pmatrix} = \begin{bmatrix} 0 & C \\ 0 & A \end{bmatrix} \begin{pmatrix} e(k) \\ z(k) \end{pmatrix} + \begin{bmatrix} 0 \\ B \end{bmatrix} (u(k) - u(k-1)) \tag{31}
$$

where $z(k)$ denotes the difference of the state variable $x(k)$, i.e., $z(k) = x(k) - x(k-1)$.

If the system shown in (31) is controllable, then a set of feedback control signals can be designed to make the system stable, i.e.,

$$
u(k) - u(k-1) = -\begin{bmatrix} F_1 & F_2 \end{bmatrix} \begin{pmatrix} e(k) \\ z(k) \end{pmatrix} \tag{32}
$$

This means that the tracking error $e(k)$ is stable. Therefore, the system is then able to track the reference input signal with zero steady-state error. Integrating (33), the feedback control signal inside the system is obtained as

$$
u(k) = -F_1 \sum_{i=1}^{k} e(k) - F_2 x(k) \tag{33}
$$

Since the state $x(k)$ of the system cannot be obtained directly, the optimal state estimate $\hat{x}(k|k)$ is obtained by an improved Kalman filtering method and used to replace the true state $x(k)$ in (34).

## 4. Simulation Results

In this section, a practical simulation is carried out to demonstrate the effectiveness of the proposed optimal state estimation method and recovery control strategy.

### 4.1. Simulation Setup

Consider a four-wheeled Omnidirectional Mobile Robot (OMR) with the system parameters shown below [14]:

$$
A = \begin{bmatrix} -0.1759 & 8.0754 \times 10^{-4} & 0.0000 \\ 8.0754 \times 10^{-4} & -0.1759 & 0.0000 \\ 0.0000 & 0.0000 & -0.0675 \end{bmatrix}, B = \begin{bmatrix} 0.0299 & 0.0299 & 0.0299 & 0.0299 \\ 0.0299 & -0.0299 & 0.0299 & -0.0299 \\ -0.0887 & -0.0887 & 0.0887 & 0.0887 \end{bmatrix}
$$

It is assumed that the mobile robot contains three independent sensors, each of which can independently measure partial or complete state information. So, the matrix of system output parameters corresponding to each sensor is given as

$$
C^{s_1} = \begin{bmatrix} 0.9 & 0 & 0 \\ 0 & 0.9 & 0 \\ 0 & 0 & 0.9 \end{bmatrix}, C^{s_2} = \begin{bmatrix} 0.8 & 0 & 0 \\ 0 & 0.8 & 0 \\ 0 & 0 & 0.8 \end{bmatrix}, C^{s_3} = \begin{bmatrix} 0.6 & 0 & 0 \\ 0 & 0.6 & 0 \\ 0 & 0 & 0.6 \end{bmatrix}
$$

The state variables $x = \begin{bmatrix} \dot{x}, \dot{y}, \dot{\theta} \end{bmatrix}^T$ of the OMR are the $X$-axis and $Y$-axis travel velocity and rotation angular velocity in the robot coordinate system. The process noise is set to $w(k) \sim N(0, 0.001)$ and the measurement noise of the three sensors is set to $v^1(k) \sim N(0, 0.008)$, $v^2(k) \sim N(0, 0.005)$, and $v^3(k) \sim N(0, 0.003)$ respectively.

In the recovery control, the feedback matrix of (34) is set to:

$$
F = \begin{bmatrix} F_1 & F_2 \end{bmatrix} = \begin{bmatrix} 0.8721 & 0.8356 & -0.2802 & 4.5969 & 4.4453 & -1.8020 \\ 0.8660 & -0.8306 & -0.2839 & 4.5894 & -4.4127 & -1.8089 \\ 0.8760 & 0.8369 & 0.2956 & 4.6138 & 4.4478 & 1.8594 \\ 0.8700 & -0.8293 & 0.2919 & 4.6064 & -4.4102 & 1.8526 \end{bmatrix}
$$

On the monitoring side, the expected movement strategy is set to drive forward in a straight line with a speed of $\dot{x}(k) = 0.5$ from the initial state $x(k) = [0,0,0]^T$, during

which the output of the sensor $s_1$ is mainly used, i.e., the sensor fusion module is set to: $\Phi(y^{s_1,a}|y^{s_2,a}|y^{s_3,a}) = 0.9y^{s_1,a} + 0.05y^{s_2,a} + 0.05y^{s_3,a}$. Without considering the attack, the normal state of the mobile robot is shown in Figure 4.

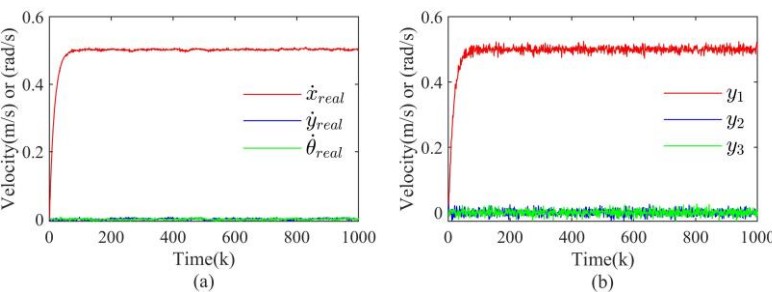

**Figure 4.** The normal state and fused measurement output of the OMR when no attack is added: (**a**) is the normal state; (**b**) is the fused measurement output.

**Remark 4.** *In all figures of this paper, the asterisk superscript (i.e., \*) indicates that the results obtained using the method proposed in this paper were used. In contrast, the absence of the asterisk superscript indicates that the results obtained by the method proposed in this paper were not used.*

*4.2. Results Discussion*

4.2.1. Zero-Dynamics Attack

Zero-dynamics attack is mainly stealthy by designing the attack signal $a_u(k)$, and there is no direct attack signal to attack the measurement channels. Therefore, in the simulation, the measurement output matrix $C^{s_1}$ of the sensor $s_1$ is used to solve (5) and assume that the sensor $s_1$ does not have access to the rotational state information. The parameters related to the zero-dynamics attack design in (5) are calculated as follows and injected into the actuation channels at $k > 300$.

$$v = 1.008, g = [-0.0803; -0.0803; 0.0803; 0.0803]$$

The real state, estimated state, and fused measurement output of the OMR with and without taking the proposed approach in this paper after the injection attack are given in Figure 5. From Figure 5b,d, it can be found that the real state of the OMR can be basically estimated using the method proposed in this paper, and the fused measurement output can also show the real state of the system very well.

After detecting the attack and estimating the correct state of the OMR, recovery control is introduced, as shown in Figure 6, which shows the system after adding recovery control. From the figure, by taking recovery control, the rotation angular velocity of the OMR can be adjusted so that it will no longer rotate.

4.2.2. Covert Attack

The main purpose of the covert attack is the attack on the actuation channels. Therefore, at the simulation moment $k > 300$, the attack target is set to $x_a(k) = [1,0,0]^T$, i.e., after injecting the attack, the resulting state of the OMR is straight ahead with a speed of 1 m/s. Also, to satisfy the stealthy condition, the attack signal is added to the sensor $s_1$ according to (6) without attacking the other two sensors.

The real state, estimated state, and fused measurement output of the OMR with and without taking the proposed approach in this paper after the injection attack are given in Figure 7. From Figure 7a, it is found that a better estimation of the system state can be achieved when the state of the OMR is estimated by adopting the method proposed in this paper. In this case, the actual state is 1 m/s, while the average value of the estimated state is 1.07 m/s. In addition, Figure 7b also demonstrates that the fused measurement output after detection is also valid.

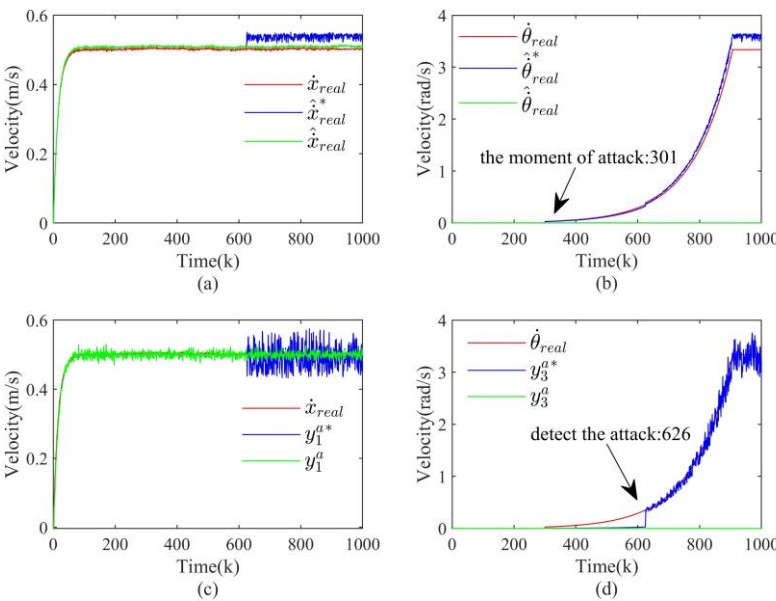

**Figure 5.** The real state, estimated state, and fused measured output of the OMR with and without taking the proposed approach in this paper after the injection attack: (**a**) is the real state and estimated state of the *X*-axis velocity; (**b**) is the real state and estimated state of the rotation angular velocity; (**c**) is the fused measurement output of the *X*-axis velocity; (**d**) is the fused measurement output of the rotation angular velocity.

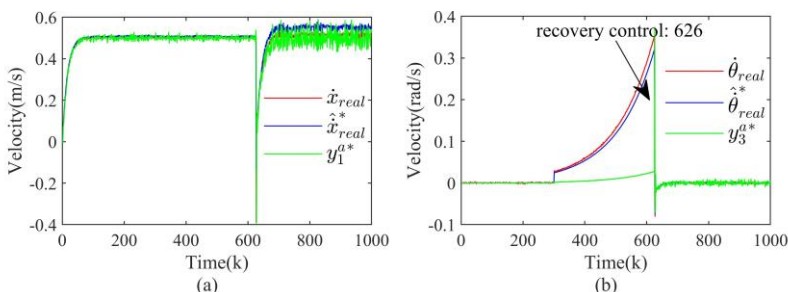

**Figure 6.** The situation of the OMR after adding recovery control: (**a**) is the situation of the *X*-axis velocity; (**b**) is the situation of the rotation angular velocity.

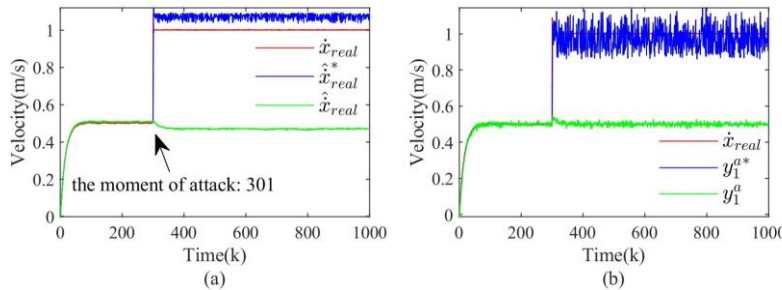

**Figure 7.** The real state, estimated state, and fused measured output of the OMR with and without taking the proposed approach in this paper after the injection attack: (**a**) is the real state and estimated state of the *X*-axis velocity; (**b**) is the fused measurement output of the *X*-axis velocity.

Figure 8 illustrates the change in the system state after taking recovery control at $k > 600$. The results show that the OMR can restore the normal driving state by adopting the recovery control strategy.

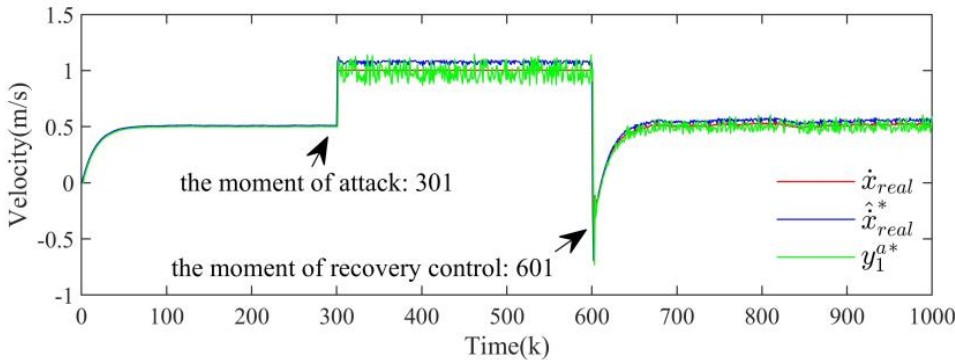

**Figure 8.** The change in the system state after taking recovery control.

### 4.2.3. Replay Attack

According to the production conditions of the replay attack, the attacked sensor is $s_1$, the target $x_a(k) = [0, 0.3, 0]^T$. The attack strategy is described as follows: collect the output data of the sensor $s_1$ for $100 < k < 300$, inject the attack at $k > 300$, continuously overwrite the output data of the sensor $s_1$ with the previously collected data, and simultaneously calculate $a_u(k)$ injected into the actuation channels according to the attack target $x_a(k)$.

The real state, estimated state, and fused measurement output of the OMR with and without taking the proposed approach in this paper after the injection attack are given in Figure 9. Figure 10 illustrates the change in the system state after taking recovery control at $k > 600$. The results show that the OMR can restore the normal driving state by adopting the recovery control strategy.

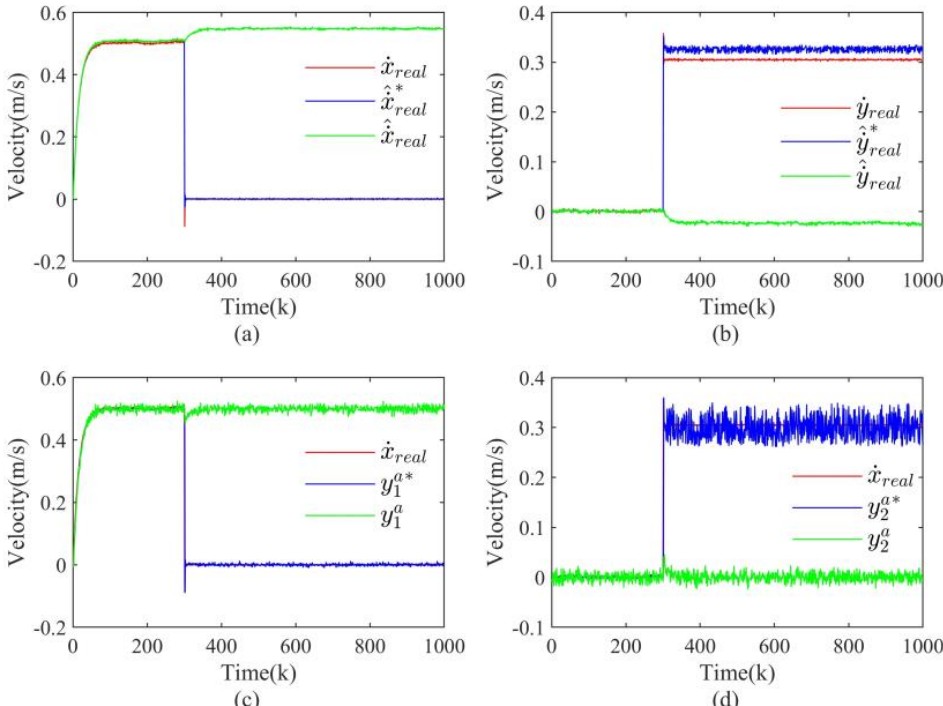

**Figure 9.** The real state, estimated state, and fused measured output of the OMR with and without taking the proposed approach in this paper after the injection attack: (**a**) is the real state and estimated state of the *X*-axis velocity; (**b**) is the real state and estimated state of the *Y*-axis velocity; (**c**) is the fused measurement output of the *X*-axis velocity; (**d**) is the fused measurement output of the *Y*-axis velocity.

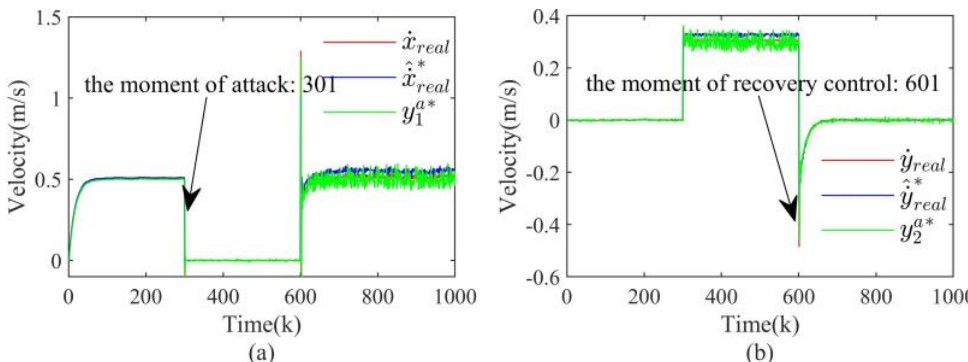

**Figure 10.** The situation of the OMR after adding recovery control: (**a**) is the situation of the *X*-axis velocity; (**b**) is the situation of the *Y*-axis velocity.

## 5. Conclusions

This paper introduces a comprehensive framework that addresses the security defense of CPSs. The framework encompasses three key components: attack detection, secure state estimation, and recovery control. The proposed processes hold significant importance in enhancing the security of CPSs. To commence, an analysis is conducted on the structural characteristics of CPSs, resulting in the development of a state-space model that encompasses sensor attacks, actuator attacks, and process attacks. A comprehensive description of the three different categories of stealthy attacks is provided. Next, we employ an existing attack detection method that is based on improved residuals to effectively detect these stealthy attacks. Building upon the obtained detection results, this study proposes an optimal state estimation method that utilizes an improved Kalman filtering approach. This method allows for precise estimation of the true state of the system. Finally, the utilization of internal model control is proposed to enhance system recovery control through the utilization of optimal state estimation. Simulation verification showcases the exceptional efficacy of the proposed methods in achieving secure state estimation and recovery control. In our future research endeavors, our objective is to tackle the nonlinearity inherent in CPSs and investigate more advanced attack behaviors to further strengthen the defense mechanisms of CPSs. Additionally, the integration of artificial intelligence and reinforcement learning methodologies into the stage of system recovery presents promising prospects for enhancing the overall security of the system.

**Author Contributions:** Conceptualization, L.X., B.Y. and Z.L.; Data curation, B.Y.; Formal analysis, L.X.; Funding acquisition, Z.L.; Methodology, B.Y., L.X. and Z.L.; Writing—original draft, B.Y.; Writing—review and editing, B.Y., L.X. and Z.L. All authors have read and agreed to the published version of the manuscript.

**Funding:** This research was funded by the National Natural Science Foundation of China grant number 52232013.

**Data Availability Statement:** Data are contained within the article.

**Conflicts of Interest:** The authors declare no conflict of interest.

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
