# Peer review of "Research on Secure State Estimation and Recovery Control for CPS under Stealthy Attacks"

_actuators, doi:10.3390/act12110427_

Round 1

Reviewer 1 Report

Comments and Suggestions for Authors

The paper presents a framework of recovering a cyber-physical system (CPS) with linear dynamics under stealth attacks. The system model is given as a discrete-time linear difference equation with all parameters specified explicitly. The input from attackers is defined by a set of attacking variables whose values are given by 3 types of stealth attacks, i.e., zero-dynamics attack, covert attack and replay attack. The detection of an attack is performed by comparing the corresponding values from the observers on the monitoring side and plant side respectively. When an attack is detected, an optimal state estimation is computed from the improved Kalman filtering method. To recover the CPS, the paper simply uses internal model control based on the obtained optimal state estimation.

Strengths:

- Attack detection and only recovery of CPS is an important problem in cybersecurity. The paper addresses a significant problem.

- The optimal state estimation is interesting and technically sound.

Weaknesses:

- The system model is too explicit. In an attack detection or system recovery problem, we should not have too many assumptions on the model.

- It is often too ideal to just compare the values of the observers. How can we know that the data is securely transferred? 

- The recovery of a CPS is not that simple. Using an internal model control might ensure the stability even the attack is still on going, but it is hard to also ensure the safety.

I also think it better not to uniformly distribute the contributions in all of the 3 aspects: detection, state estimation and recovery. This paper can be focusing on state estimation, since the contribution in the other two aspects is not strong. Besides, I would like the authors to discuss the related work on real-time system recovery and stealth attack detection. There is a lot of existing work in these two topics and most of them even consider nonlinear or blackbox systems.  

Comments on the Quality of English Language

The English writing and notations should be improved. For example:

- Line 126: I am not able to find u^a(k).

- Line 261: "whenever a sensor is detected an attack" is not a correct English sentence.

Also, the writing of Section 3 should be improved. More intuitions should be provided, and if a step is from an existing work, it should be explicitly pointed out.

Author Response

Response to Reviewer 1 Comments

Thank you for allowing a resubmission of our manuscript, with an opportunity to address your comments.

Point 1: The system model is too explicit. In an attack detection or system recovery problem, we should not have too many assumptions on the model.

Response 1: Thanks for your kind suggestion. In practice, it is indeed uncommon to make extensive assumptions about the model in the context of attack detection or system recovery. Such assumptions can often lead to overly idealized outcomes. In this paper, we primarily focused on establishing a linear model for the system, without considering nonlinearity. Although there are ongoing efforts by researchers to develop nonlinear models, the use of a linear model is a common approach in CPS defense research. Additionally, our work primarily considered additive attacks and did not explicitly address multiplicative attacks. Multiplicative attacks mainly affect the system's model parameters, introducing nonlinear behavior. While there is some existing research on multiplicative attacks, it remains relatively limited compared to studies on various types of additive attacks. Therefore, our assumptions are justified within the scope of this study, and we did not make many assumptions in other aspects. However, for future investigations, it would be important to consider additional factors such as nonlinearity, multiplicative attacks, and the stochastic nature of attack mechanisms. Thank you again for your suggestion, and we will certainly take these points into consideration for further research.

Point 2: It is often too ideal to just compare the values of the observers. How can we know that the data is securely transferred?

Response 2: Thanks for your kind suggestion. In this paper, for the detection of stealthy attacks, we compare the differences between observed values at the plant side and the monitoring side. To ensure the security of observed data transmission, we mentioned adopting a specific encoding strategy known only to the targeted party. However, if the attacker gains knowledge of the encoding strategy, the integrity of the observed data transmission may become compromised. Nonetheless, such cases involve more complexity and the assurance of secure data transmission becomes challenging when dealing with sophisticated attacks. Furthermore, in our design, the encoding strategy is adaptable and can be switched, thereby significantly enhancing the security of data transmission. Thank you again for your comments, and we will consider these points for further improvement.

Point 3: The recovery of a CPS is not that simple. Using an internal model control might ensure the stability even the attack is still on going, but it is hard to also ensure the safety.

Response 3: Thanks for your kind suggestion. The issue you raised is indeed profound. In this study, our primary focus is on the recovery objective, which involves utilizing internal model control to bring the system back to its normal operating state when it deviates due to an attack. However, we don’t address the security aspect during this recovery process since the attack is still present and the system remains vulnerable. Nevertheless, this could potentially become one of the significant areas for future research – exploring strategies to enhance the system's security once an attack is detected. In the concluding remarks, we will provide some thoughts on this novel idea. Once again, we appreciate your valuable suggestions.

Point 4: I also think it better not to uniformly distribute the contributions in all of the 3 aspects: detection, state estimation and recovery. This paper can be focusing on state estimation, since the contribution in the other two aspects is not strong. Besides, I would like the authors to discuss the related work on real-time system recovery and stealth attack detection. There is a lot of existing work in these two topics and most of them even consider nonlinear or blackbox systems.

Response 4: Thanks for your kind suggestion. Based on your advice, we have reorganized the contributions of this paper, emphasizing the contribution of state estimation. We have also integrated attack detection, recovery, and security defense frameworks into one contribution. Furthermore, as per your suggestion, we have reviewed and summarized relevant works on real-time system recovery and stealthy attack detection. Additionally, we have reorganized the introduction section. We hope these revisions adequately address your concerns. Thank you for your valuable feedback.

Point 5: The English writing and notations should be improved. For example: Line 126: I am not able to find u^a(k). Line 261: "whenever a sensor is detected an attack" is not a correct English sentence. Also, the writing of Section 3 should be improved. More intuitions should be provided, and if a step is from an existing work, it should be explicitly pointed out.

Response 5: Thank you for your suggestions. In the paper, we have provided an explanation for the meaning of ${{u}^{a}}(k)$ in Line 162. ${{u}^{a}}(k)$ represents the control signal input to the plant side after the attack, specifically defined as ${{u}^{a}}(k)=u(k)+{{a}_{u}}(k)$. Regarding the sentence in Line 261 (now Line 298), we have rewritten it. Furthermore, we have restructured and rewritten the Section III, seeking assistance from native English speakers to improve the overall writing and expression throughout the entire paper.

Reviewer 2 Report

Comments and Suggestions for Authors

This paper deals with secure state estimation and recovery control of CPSs subject to stealthy attacks. An improved-residual-based method is used to detect stealthy attacks, and an optimal state estimation method is proposed to estimate the system state. Then an internal mode control scheme is introduced for recovery control of the system. The simulation results show the effectiveness of the proposed method. This paper is interesting, and the results seem correct and reasonable. However, the following comments should be considered when revised.

1.      Regarding the security of networked control systems or CPSs, related results should be mentioned in the Introduction, e.g. “Event-triggered output feedback synchronization of master-slave neural networks under deception attacks”, “A novel approach to H-infinity performance analysis of discrete-time networked systems subject to network-induced delays and malicious packet dropouts," and "Network-based modeling and proportional-integral control for direct-drive-wheel systems in wireless network environments,"

2.      Lines 122-124, it is said that “The CPS model with … is constructed as …”. To the reviewer, it should be “given as” rather than “constructed as”.

3.      In Remark 1, what are “Process attacks”, “direct attacks”, and “additive attacks” in Remark 2? Please give more explanation.

4.      In the Section 3.1, why do you say that the state estimation is optimal?

5.      Lines 332-334, it is hard to understand. Is it right that “the system shown in (32) is controllable”? How to understand the system (32) is stable? Why do you say “the tracking error e(k) is stable on the line 335?

6.      Is it possible to make some comparisons in the simulation?

7.      The paper needs to be polished largely.

Comments on the Quality of English Language

Please polish the paper by an English-speaking expert.

Author Response

Response to Reviewer 2 Comments

Thank you for allowing a resubmission of our manuscript, with an opportunity to address your comments.

Point 1: Regarding the security of networked control systems or CPSs, related results should be mentioned in the Introduction, e.g. “Event-triggered output feedback synchronization of master-slave neural networks under deception attacks”, “A novel approach to H-infinity performance analysis of discrete-time networked systems subject to network-induced delays and malicious packet dropouts," and "Network-based modeling and proportional-integral control for direct-drive-wheel systems in wireless network environments,"

Response 1: Thanks for your kind suggestion. We have carefully reviewed these research works and found that they contribute to our comprehensive review of CPS security-related work in the introduction section. Therefore, when reorganizing the content of the introduction, we have included a review and citation of the aforementioned related works to enhance the completeness of our investigation. Thank you for bringing these to our attention.

Point 2: Lines 122-124, it is said that “The CPS model with … is constructed as …”. To the reviewer, it should be “given as” rather than “constructed as”.

Response 2: Thanks for your kind suggestion. According to your advice, we have revised the relevant portions accordingly.

Point 3: In Remark 1, what are “Process attacks”, “direct attacks”, and “additive attacks” in Remark 2? Please give more explanation.

Response 3: Thanks for your kind suggestion. "Process attacks" refer to attacks that are directly applied to physical devices, causing direct damage to the physical devices. Our previous use of "direct attacks" was incorrect, and we appreciate you pointing that out. "Additive attacks" are a type of attack that involves interference or noise added to the input or output signals of a system. This attack increases the amplitude of the signal, thereby affecting the normal operation and accuracy of the system. In this paper, additive attacks are represented by equation (1). Additive attacks are a common type of attack aimed at compromising the integrity and reliability of the system. According to your advice, we have made the necessary corrections and clarified the definitions of process attacks and additive attacks in our manuscript.

Point 4: In the Section 3.1, why do you say that the state estimation is optimal?

Response 4: Thanks for your kind suggestion. In this paper, we implemented sensor attack isolation based on the detection results and utilized Kalman filtering for system state estimation. Kalman filtering assumes that the system's dynamic model and observation errors follow Gaussian distributions, which are characterized by minimum mean squared error. In the previous section, we established the assumption that measurement and process errors both satisfy Gaussian distributions. Therefore, under these assumptions, Kalman filtering can estimate the system's state in a manner that minimizes the mean squared error. Moreover, Kalman filtering takes into account the prior information of the system and the observed data, providing optimal state estimation through recursive updates. It incorporates previous state estimates and measurements while weighting and fusing new observations, thus achieving optimality. In conclusion, under the aforementioned assumptions, this paper utilizes Kalman filtering for state estimation, resulting in the optimal state estimation in terms of minimum mean squared error.

Point 5: Lines 332-334, it is hard to understand. Is it right that “the system shown in (32) is controllable”? How to understand the system (32) is stable? Why do you say “the tracking error e(k) is stable on the line 335?

Response 5: Thanks for your kind suggestion. Firstly, I apologize for the mistake in Line 370 where it should be "the system shown in (31) is controllable" rather than equation (32). In this context, the sentence "the system shown in (31) is controllable" can be properly explained. In addition, in internal model control, when a control signal is designed to stabilize the control system, as shown in equation (32), the tracking error converges within a bounded range and does not exhibit infinite growth or divergence. Therefore, we can express it as "the tracking error e(k) is stable".

Point 6: Is it possible to make some comparisons in the simulation?

Response 6: Thanks for your kind suggestion. While we appreciate your idea of including comparisons in the simulation, we have intentionally chosen not to include such comparisons in this study. Our focus is primarily on exploring a specific aspect or technique of CPS defense, rather than comparing it with other methods or approaches. By focusing solely on the proposed method and its effectiveness within the scope of our research objective, we aim to provide a comprehensive and detailed analysis of its performance. This approach allows us to provide a deeper understanding of the proposed method and its potential benefits without the need for direct comparisons. We believe that presenting the outcomes and insights gained from our research, as well as highlighting any limitations and future directions, will contribute significantly to the existing literature in this field.

Point 7: The paper needs to be polished largely. Please polish the paper by an English-speaking expert.

Response 7: Thanks for your kind suggestion. We acknowledge that the paper may require significant polishing to improve its overall quality. We appreciate your suggestions and will take them into consideration while revising the paper. We understand the importance of ensuring clarity, coherence, and proper organization in the paper. We will carefully review the manuscript to identify areas that need improvement, such as sentence structure, grammar, and logical flow. Additionally, we will pay close attention to addressing any existing inconsistencies or ambiguities in the content. Our aim is to enhance the readability and comprehensibility of the paper, thereby improving its overall presentation. We will work diligently to refine the writing style, use appropriate technical terminology, and provide concise explanations where necessary. We value your input and are committed to making the necessary revisions to ensure the highest possible quality of the paper. Your guidance and suggestions are greatly appreciated, and we will strive to address all the concerns raised during the revision process.

Round 2

Reviewer 1 Report

Comments and Suggestions for Authors

I would like to thank the authors for improving the paper. The description of the technical contributions is clearer. However, I would still like the authors to address the following aspect:

- Whenever a new method is introduced, please provide the references of the existing results based on which the new method is developed.

It can better help the readers to evaluate the contributions.

Comments on the Quality of English Language

Th English writing is improved. 

Author Response

Response to Reviewer 1 Comments

Thank you for allowing a resubmission of our manuscript, with an opportunity to address your comments.

Point 1: Whenever a new method is introduced, please provide the references of the existing results based on which the new method is developed.

Response 1: Thanks for your kind suggestion. Based on your advice, we have added references to the new method mentioned in our manuscript. For example, in section 2.3, when referring to the use of a stealthy attacks detector, we have cited reference [31]. In section 3.2, when discussing related work on internal model control (IMC), we have cited reference [34]. Lastly, in the simulation setup, the model parameters were referenced from literature [14]. The above-mentioned content in our manuscript is further research or application on the basis of the corresponding literature works. We have followed your advice and incorporated the appropriate citations.

Reviewer 2 Report

Comments and Suggestions for Authors

All the comments have been addressed.

Author Response

Thank you very much for your recognition of our work and the modifications made. We sincerely appreciate you taking the time to review our paper. Wishing you good health and success in your research endeavors!